# Observing one-divalent-metal-ion-dependent and histidine-promoted His-Me family I-PpoI nuclease catalysis *in crystallo*

Caleb Chang, Grace Zhou, Yang Gao*

Department of Biosciences, Rice University, Houston, United States

**Abstract** Metal-ion-dependent nucleases play crucial roles in cellular defense and biotechnological applications. Time-resolved crystallography has resolved catalytic details of metal-ion-dependent DNA hydrolysis and synthesis, uncovering the essential roles of multiple metal ions during catalysis. The histidine-metal (His-Me) superfamily nucleases are renowned for binding one divalent metal ion and requiring a conserved histidine to promote catalysis. Many His-Me family nucleases, including homing endonucleases and Cas9 nuclease, have been adapted for biotechnological and biomedical applications. However, it remains unclear how the single metal ion in His-Me nucleases, together with the histidine, promotes water deprotonation, nucleophilic attack, and phosphodiester bond breakage. By observing DNA hydrolysis *in crystallo* with His-Me I-PpoI nuclease as a model system, we proved that only one divalent metal ion is required during its catalysis. Moreover, we uncovered several possible deprotonation pathways for the nucleophilic water. Interestingly, binding of the single metal ion and water deprotonation are concerted during catalysis. Our results reveal catalytic details of His-Me nucleases, which is distinct from multi-metal-ion-dependent DNA polymerases and nucleases.

**\*For correspondence:**
yg60@rice.edu

**Competing interest:** The authors declare that no competing interests exist.

## eLife assessment

Chang et al. have investigated the catalytic mechanism of I-PpoI nuclease, a one-metal-ion dependent nuclease, by time-resolved X-ray crystallography using soaking of crystals with metal ions under different pH conditions. This **convincing** study revealed that I-PpoI catalyzes the reaction process through a single divalent cation. The study uncovers **important** details of the roles of the metal ion and the active site histidine in catalysis.

## Introduction

Mg$^{2+}$-dependent nucleases play fundamental roles in DNA replication and repair (*Kao and Bambara, 2003*; *Shen et al., 2005*; *Marti and Fleck, 2004*; *Mimitou and Symington, 2009*), RNA processing (*Patel and Steitz, 2003*; *Abelson et al., 1998*; *Chu and Rana, 2007*; *Moore and Proudfoot, 2009*), as well as immune response and defense (*James et al., 1996*; *Sorek et al., 2008*; *Tock and Dryden, 2005*). Moreover, they are widely employed for genome editing in biotechnological and biomedical applications (*Adli, 2018*; *Carroll, 2014*). These nucleases are proposed to cleave DNA through a SN$_2$-type reaction, in which a water molecule, or sometimes, a tyrosine side chain (*Grindley et al., 2006*), initiates the nucleophilic attack on the scissile phosphate with the help of metal ions (*Yang, 2011*). Metal ions can orient and stabilize the binding of the negatively charged nucleic acid backbone (*Chen et al., 2017*), promoting proton transfer, nucleophilic attack, and stabilization of the transition

state. As a highly varied family of enzymes, $Mg^{2+}$-dependent nucleases can be broadly categorized by the number of metal ions captured in their active site. So far, $Mg^{2+}$-dependent nucleases with one and two metal ions have been observed. In the two-$Mg^{2+}$-ion-dependent nuclease, a metal ion binds on the leaving group side of the scissile phosphate ($Me^{2+}_B$) while the other binds on the nucleophile side ($Me^{2+}_A$). In one-metal-ion-dependent nucleases, only the metal ion corresponding to the B site in two-metal-ion-dependent nucleases is present (*Yang, 2008*). Moreover, transiently bound metal ions have been identified in the previously thought two-metal-ion RNaseH (*Samara and Yang, 2018*) and EndoV nucleases (*Wu et al., 2019*) via time-resolved X-ray crystallography, proposed to play key roles in various stages of their catalysis. Similarly, catalysis in the one-metal-ion-dependent APE1 nuclease has been observed *in crystallo*, but mechanistic details regarding its metal-ion dependence have not been thoroughly explored (*Freudenthal et al., 2015*; *Whitaker et al., 2018*; *Dupureur, 2010*). There also exist three-$Zn^{2+}$-dependent nucleases, with two $Zn^{2+}$ binding in the A- and B-equivalent positions, while the third $Zn^{2+}$ coordinating the sp oxygen on the nucleophile side of the scissile phosphate (*Garcin et al., 2008*). However, it remains unclear how the single metal ion in one-metal-ion-dependent nucleases is capable of aligning the substrate, promoting deprotonation and nucleophilic attack, and stabilizing the pentacovalent transition state.

A large subfamily of one-metal-ion-dependent nucleases consist of histidine-metal (His-Me) nucleases that perform critical tasks in biological pathways such as apoptosis (*Arnoult et al., 2003*; *Lin et al., 2016*), extracellular defense (*Cheng et al., 2002*; *Hsia et al., 2004*), intracellular immunity (CRISPR–Cas9) (*Doudna and Charpentier, 2014*; *Ran et al., 2013*), and intron homing (homing endonuclease) (*Chevalier and Stoddard, 2001*; *Stoddard, 2005*). Despite sharing poor sequence homology, the structural cores and active sites of His-Me nucleases are highly conserved and thus are proposed to catalyze DNA hydrolysis through a similar mechanism (*Yang, 2011*; *Stoddard, 2005*; *Wu et al., 2020*). Surrounded by two beta sheets and an alpha helix, the His-Me nucleases active sites all contain a single metal ion, a histidine, and an asparagine (*Figure 1A*, *Figure 1—figure supplement 1A*, and *Figure 1—figure supplement 2*). The asparagine residue helps to stabilize metal ion binding whereas the strictly conserved histidine is suggested to deprotonate a nearby water for nucleophilic attack toward the scissile phosphate (*Galburt et al., 1999*; *Pommer et al., 2001*; *Li et al., 2003*; *Shen et al., 2004*). To this day, the dynamic reaction process of DNA hydrolysis by His-Me nucleases has never been visualized, and the mechanism of single metal-ion-dependent and histidine-promoted catalysis remains unclear. Emerging genome editing and disease treatment involving CRISPR–Cas9 emphasize the importance of understanding the catalytic mechanism of DNA hydrolysis by His-Me nucleases (*Schwank et al., 2013*; *Sharma et al., 2021*; *Khan et al., 2016*). Recent crystal and cryo-electron microscope structures of Cas9 have captured the His-Me family Cas9 histidine–asparagine–histidine (HNH) active site engaged with DNA before and after cleavage (*Sun et al., 2019*; *Zhang et al., 2020*; *Zhu et al., 2019*; *Das et al., 2023*). However, due to the relatively low resolution and the static nature of the structures, key catalytic details, such as metal ion dependence, transition-state stabilization, and alternative deprotonation pathways, remain elusive. Moreover, the large size and multiple conformational checkpoints during Cas9 catalysis (*Sun et al., 2019*; *Zhang et al., 2020*; *Zhu et al., 2019*; *Das et al., 2023*) hinder *in crystallo* observation of Cas9 catalysis.

I-PpoI is a well-characterized intron-encoded homing endonuclease member of the *physarum polycephalum* slimemold. By 1999, Stoddard and colleagues were able to capture intermediate structures of I-PpoI complexed with DNA before and after product formation (*Galburt et al., 1999*; *Flick et al., 1998*). We herein employed I-PpoI as a model system and applied time-resolved crystallography to observe the catalytic process of His-Me nuclease. By determining over 40 atomic resolution structures of I-PpoI during its reaction process, we show that one and only one divalent metal ion is involved in DNA hydrolysis. Moreover, we uncover several possible deprotonation pathways for the nucleophilic water. Notably, metal ion binding and water deprotonation are highly concerted during catalysis. Our findings provide mechanistic insights into one-metal-ion-dependent nucleases, enhancing future design and engineering of these enzymes for emerging biomedical applications.

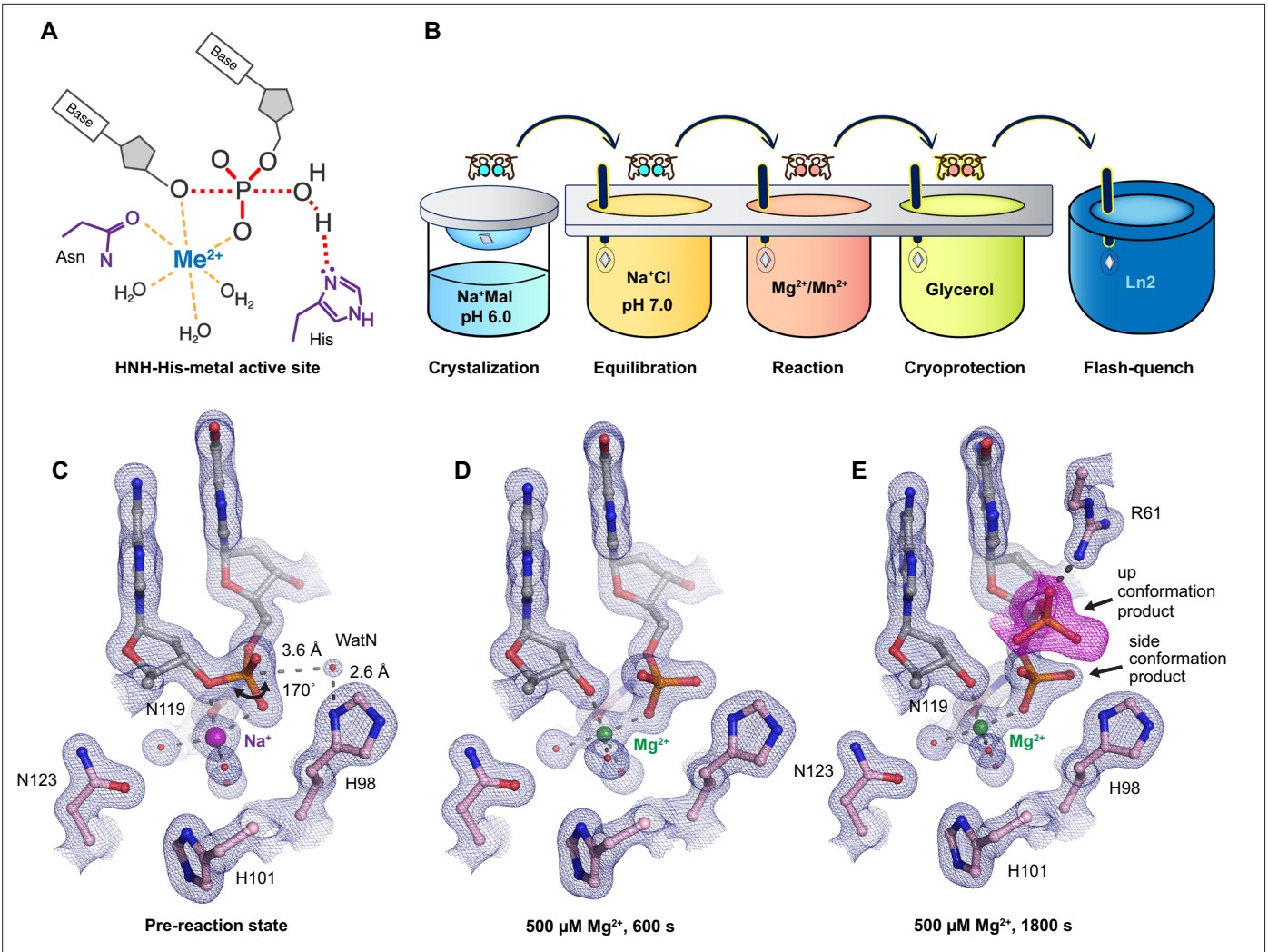

**Figure 1.** Observing histidine-metal (His-Me) family I-PpoI catalyze DNA hydrolysis *in crystallo*. (**A**) Model of one-metal-ion-dependent and histidine-promoted His-Me enzyme catalysis and transition-state stabilization. (**B**) Metal ion soaking setup for *in crystallo* observation of DNA hydrolysis with I-PpoI. Structural intermediates of I-PpoI *in crystallo* DNA cleavage showcasing the pre-reaction state in (**C**) and product states in (**D**) and (**E**). The $2F_o - F_c$ map for $Me^{2+}$, DNA, waters (red spheres), and catalytic residues (blue) was contoured at 2.0 σ (σ values represent root mean square (r.m.s.) density values (later refered to as r.m.s.d.)). (**E**) The $F_o - F_c$ omit map for the up conformation of the product (violet) was contoured at 3.0 σ.

The online version of this article includes the following figure supplement(s) for figure 1:

**Figure supplement 1.** Overall structure and catalytic core of homing endonuclease I-PpoI.

**Figure supplement 2.** Active sites of histidine-metal (His-Me) superfamily nucleases.

**Figure supplement 3.** Establishing I-PpoI for *in crystallo* studies.

## Results

### Preparation of the I-PpoI system for *in crystallo* DNA hydrolysis

We sought to implement I-PpoI for *in crystallo* metal ion soaking (*Figure 1B*), which has been successful in elucidating the catalytic mechanisms of DNA polymerases (*Freudenthal et al., 2013*; *Gao and Yang, 2016*; *Nakamura et al., 2012*; *Vyas et al., 2015*; *Chim et al., 2021*; *Gregory et al., 2021*), nucleases (*Samara and Yang, 2018*; *Wu et al., 2019*; *Freudenthal et al., 2015*; *Whitaker et al., 2018*), and glycosylase (*Demir et al., 2023*). First, a complex of I-PpoI and a palindromic DNA was crystalized at pH 6 with 0.2 M sodium malonate. Similar to previous studies, a dimer of I-PpoI was found in the asymmetric unit, with both active sites engaged for catalysis and DNA in the middle bent by 55° (*Figure 1—figure supplement 1A*). The two I-PpoI molecules were almost identical and thus served as internal controls for evaluating the reaction process (*Figure 1—figure supplement*

*1B*). Within each I-PpoI active site, a water molecule (nucleophilic water, WatN) existed 3.6 Å from the scissile phosphate, near the imidazole side chain of His98. On the leaving group side of the scissile phosphate, the metal exhibited an octahedral geometry, being coordinated by three water molecules, two oxygen atoms from the scissile phosphate, and the conserved asparagine (*Figure 1C*).

Next, we removed malonate in the crystallization buffer, which may chelate metal ions and hinder metal ion diffusion (*Deerfield et al., 1991*), by equilibrating the crystals in 200 mM NaCl buffer at pH 6, 7, or 8 for 30 min. The diffraction quality of the crystals was not affected during the soaking. The structures of I-PpoI equilibrated at pH 6, 7, or 8 in the presence of 200 mM NaCl showed no signs of product formation and appeared similar to the malonate and previous reported I-PpoI structures (*Galburt et al., 1999*; *Flick et al., 1998*; *Figure 1—figure supplement 3A*). To confirm if a monovalent metal ion binds in the pre-reaction state, we soaked the I-PpoI crystals in buffer containing Tl⁺ and detected anomalous electron density at the metal ion-binding site without product formation (*Figure 1—figure supplement 3B*). Our results support that the monovalent metal ion can bind within the active site without initiating reaction, in corroboration with our biochemical assays (*Figure 1—figure supplement 3C*).

## Witnessing DNA hydrolysis *in crystallo* by I-PpoI

To initiate the chemical reaction *in crystallo*, we transferred the I-PpoI crystals equilibrated in NaCl buffer to a reaction buffer with 500 µM $Mg^{2+}$ (*Figure 1B*). After 600 s soaking, we saw a significant negative $F_o − F_c$ peak on the leaving group side of the scissile phosphate atom as well as a positive $F_o − F_c$ peak on the other side (*Figure 1—figure supplement 3D*), indicating that DNA hydrolysis was occurring *in crystallo*. After DNA cleavage, the newly generated phosphate group shifted 1 Å toward the WatN (*Figure 1D*). Furthermore, an additional product state was observed after soaking the crystals in 500 µM $Mg^{2+}$ for 600 and 1800 s, at which the newly formed phosphate shifted 3 Å away from

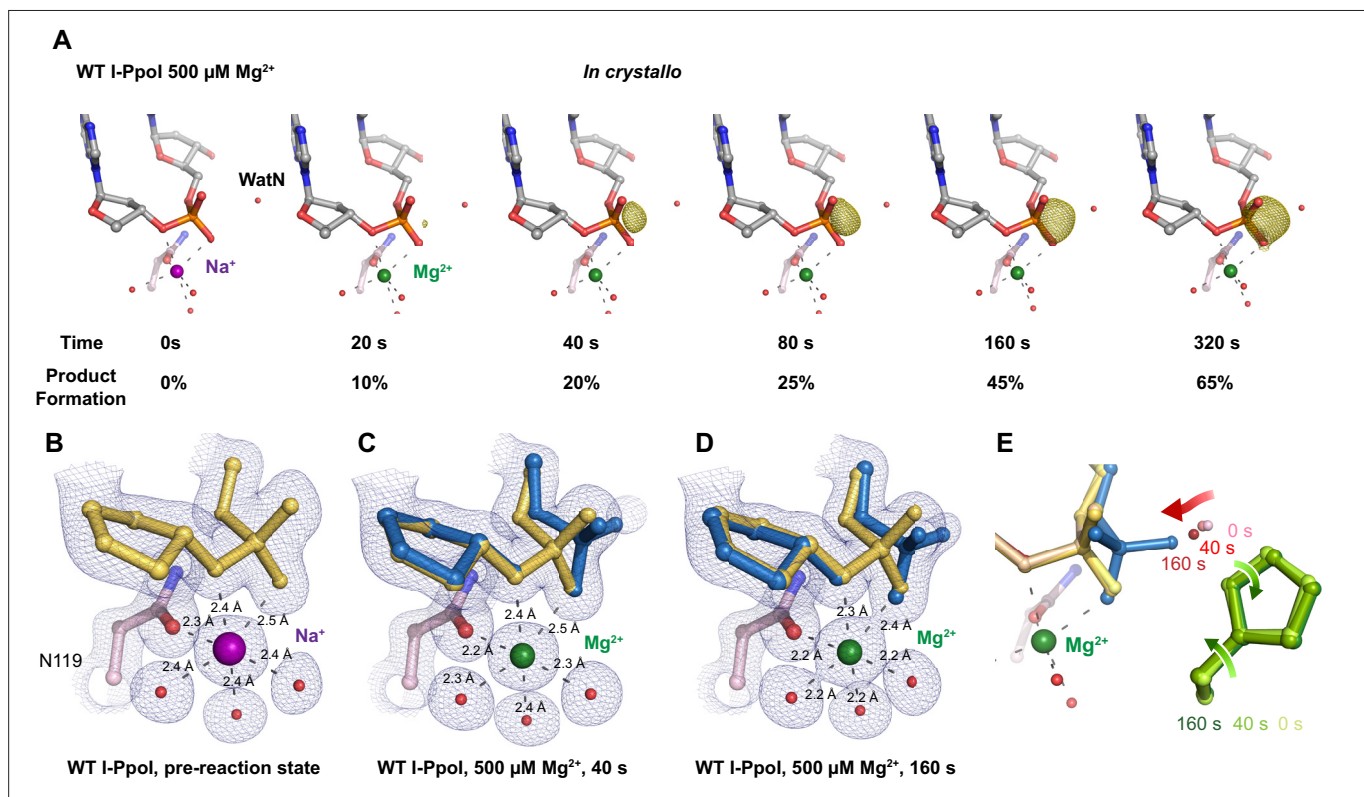

**Figure 2.** *In crystallo* DNA hydrolysis by I-PpoI. (**A**) Structures of I-PpoI during *in crystallo* catalysis after 500 µM $Mg^{2+}$ soaking for 0, 20, 40, 80, 160, and 320 s. The $F_o − F_c$ omit map for the product phosphate (green) was contoured at 3.0 σ. I-PpoI complexes featuring metal ion coordination when bound with Na⁺ in (**B**) $Mg^{2+}$ in the earlier time point of the reaction process in (**C**), and $Mg^{2+}$ in the later time point of the reaction process in (**D**). The $2F_o − F_c$ map for Me²⁺, DNA, waters (red spheres), and catalytic residues (blue) was contoured at 2.0 σ. (**E**) Alignment of the WatN and rotation in His98 during I-PpoI reaction.

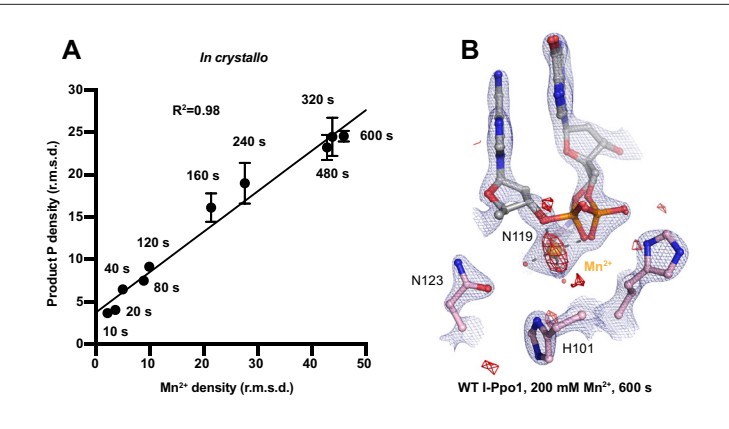

**Figure 3.** Detection of Me²⁺ binding during DNA hydrolysis *in crystallo*. (**A**) Correlation ($R^2$) between the newly formed phosphate and Mn²⁺ binding at pH 6. The points represent the mean of duplicate measurements for the electron density of the reaction product phosphate within two I-PpoI molecules in the asymmetric unit while the error bars represent the standard deviation. (**B**) Additional Mn²⁺-binding sites were not detected in the I-PpoI active site after 10 min soaking in 200 mM Mn²⁺. The $2F_o - F_c$ map for Me²⁺, DNA, waters (red spheres), and catalytic residues (blue) was contoured at 2.0 σ. The anomalous map for Mn²⁺ was collected at X-ray wavelength of 0.9765 Å and contoured at 3.0 σ.

The online version of this article includes the following figure supplement(s) for figure 3:

**Figure supplement 1.** Additional metal ions are not required for DNA hydrolysis by I-PpoI.

**Figure supplement 2.** Environment of the transient Me²⁺ in nucleases.

the metal ion to form a hydrogen bond with Arg61 (*Figure 1E*). Our results confirm that the implementation of I-PpoI with *in crystallo* Me²⁺ soaking is feasible for observing the I-PpoI catalytic process and dissecting the mechanism of His-Me nucleases.

With an established *in crystallo* reaction system, we next monitored the reaction process by soaking I-PpoI crystals in buffer containing 500 µM Mg²⁺ pH 7, for 10–1200 s. Density between the reactant phosphate and nucleophilic water increased along with longer soaking time, indicating the generation of phosphate products (*Figure 2A*). During the reaction, the coordination distances of the metal ion ligands decreased 0.1–0.2 Å when Mg²⁺ exchanged with Na⁺, which is consistent with their preferred geometry (*Cowan, 2002*; *Figure 2B, C*). At reaction time 160 s, 45% of product had been generated (*Figure 2D*), which later plateaued to 65% at 300 s. During the reaction process, the sugar ring of the reactant and product DNA that resides around 3 Å away from the scissile phosphate, remained in a C3'-endo conformation. As Mg²⁺ soaking time increased from 10 to 160 s, we observed the WatN approaching the scissile phosphate (*Figure 2E*). At the same time, the conserved His98 sidechain proposed to deprotonate the WatN was slightly rotated.

## A single divalent metal ion was captured during DNA hydrolysis

Throughout the reaction process with Mg²⁺, we found that the electron density for the Mg²⁺ metal ion strongly correlated ($R^2 = 0.97$) with product formation (*Figure 3—figure supplement 1A, C*), which may suggest that saturation of this metal ion site is required and sufficient for catalysis. However, Mg²⁺'s similar size to Na⁺ makes it suboptimal for quantifying metal ion binding. To thoroughly investigate metal ion dependence, we repeated the *in crystallo* soaking experiment with Mn²⁺, which is more electron rich and can be unambiguously assigned based on its electron density and anomalous signal. With 500 µM Mn²⁺ in the reaction buffer, we found that the reaction process and the product conformation were similar to that for Mg²⁺. After 160 s Mn²⁺ soaking, clear anomalous signal was present at the metal ion-binding site, confirming the binding of Mn²⁺ (*Figure 3—figure supplement 1F*). For crystal structures of I-PpoI with partial product formation, Mn²⁺ signal at the metal ion site correlated with product formation with a $R^2$ of 0.98 (*Figure 3A* and *Figure 3—figure supplement 1B*). To further search if additional and transiently bound divalent ions participate in the reaction, we soaked the I-PpoI crystals in high concentration of Mn²⁺ (200 mM) for 600 s, at which 80% product formed within

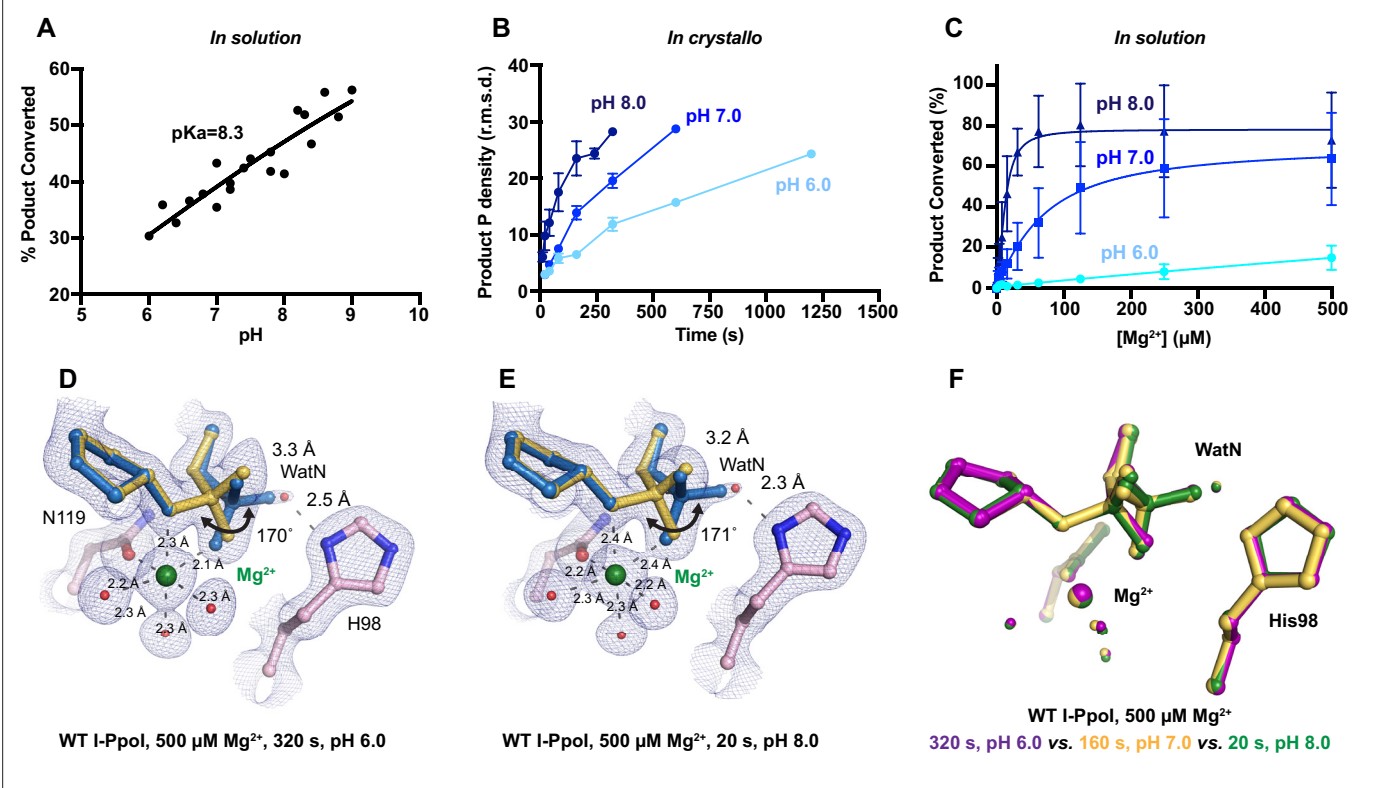

**Figure 4.** Effect of pH on I-PpoI DNA hydrolysis. (**A**) DNA hydrolysis by WT I-PpoI with increasing pH in solution. (**B**) DNA hydrolysis by WT I-PpoI *in crystallo* at pH 6, 7, and 8. The points represent the mean of duplicate measurements for the electron density of the reaction product phosphate after a period of $Mg^{2+}$ soaking within two I-PpoI molecules in the asymmetric unit. The error bars represent the standard deviation. (**C**) The effect of pH on metal ion dependence in solution. The points represent the mean of triplicate measurements for the percentage of cleaved DNA product while the error bars represent the standard deviation. (**D**) Structure of I-PpoI *in crystallo* DNA hydrolysis at pH 6 after 320 s of 500 µM $Mg^{2+}$ soaking. (**E**) Structure of I-PpoI *in crystallo* DNA hydrolysis at pH 8 after 20 s of 500 µM $Mg^{2+}$ soaking. (**D, E**) The $2F_o - F_c$ map for $Me^{2+}$, DNA, waters (red spheres), and catalytic residues (blue) was contoured at 2.0 σ. (**F**) Structural comparison of the active site after 500 µM $Mg^{2+}$ soaking for 320 s at pH 6 (violet), 160 s at pH 7 (yellow), and 20 s at pH 8 (green).

the active site. However, apart from the single metal ion-binding site, we do not detect anomalous signal for additional $Mn^{2+}$, despite such high concentration of $Mn^{2+}$ (***Figure 3B*** and ***Figure 3—figure supplement 2***). The strong correlation between product phosphate formation with $Mn^{2+}$ binding and the absence of additional anomalous density peaks in heavy $Mn^{2+}$ soaked crystals suggest that one and only one divalent metal ion is involved in I-PpoI DNA hydrolysis.

To examine if additional monovalent metal ions participate during DNA hydrolysis, we titrated $Tl^+$, which can replace monovalent $Na^+$ or $K^+$ and yields anomalous signal (***Auffinger et al., 2016***; ***Kiser et al., 2009***), at up to 100 mM concentration in the biochemical assay. Apart from precipitation that occurred at 50 and 100 mM $Tl^+$, we found that product conversion remained unaffected with increasing $Tl^+$ in solution (***Figure 3—figure supplement 1G***). Similarly, increasing $Na^+$ concentration in solution did not increase product conversion (***Figure 3—figure supplement 1H***). In addition, only one $Tl^+$ was detected by its anomalous signal in the pre-reaction state (***Figure 1—figure supplement 3B***). The biochemical and structural results indicate that additional monovalent metal ion may not be necessary for DNA hydrolysis. However, crystal deterioration limited us from soaking the I-PpoI crystals in high concentration of $Tl^+$ for *in crystallo* reaction.

## pH dependence of I-PpoI DNA cleavage

During DNA hydrolysis, deprotonation of the WatN is required for the nucleophilic attack and phosphodiester bond breakage. This proton transfer has been proposed to be mediated by the highly conserved His98 (***Galburt et al., 1999***; ***Flick et al., 1998***) that lies within 3 Å from the WatN. We speculated that the ability of His98 to activate the nearby WatN and mediate proton transfer would

be affected by pH. The DNA cleavage assay revealed that I-PpoI cleavage activity increased with pH with a pKa of 8.3 (*Figure 4A*), which is much higher than the pKa of histidine, but the histidine pKa and the proton transfer process may be affected by the active site environment. To explore how pH affects the active site configuration and catalysis, we conducted *in crystallo* soaking experiments with $Mg^{2+}$ at pH 6 and 8 in addition to the pH 7 data series in *Figure 2*. Consistent with in solution experiments, higher pH resulted in faster product formation *in crystallo* (*Figure 4B*), indicating that metal ion binding may also be affected by pH. To test this, we performed $Mg^{2+}$ titration at different pH in solution. Our results showed that over 100 times higher concentration of $Mg^{2+}$ was needed to yield 50% product in pH 6 versus 8 (*Figure 4C*), confirming that pH affects metal-ion-dependent I-PpoI catalysis. Likewise, the reactions *in crystallo* at higher pH reached 50% product formation at shorter soaking times (320 s at pH 6, 160 s at pH 7, and 80 s at pH 8, respectively). Interestingly, the crystal

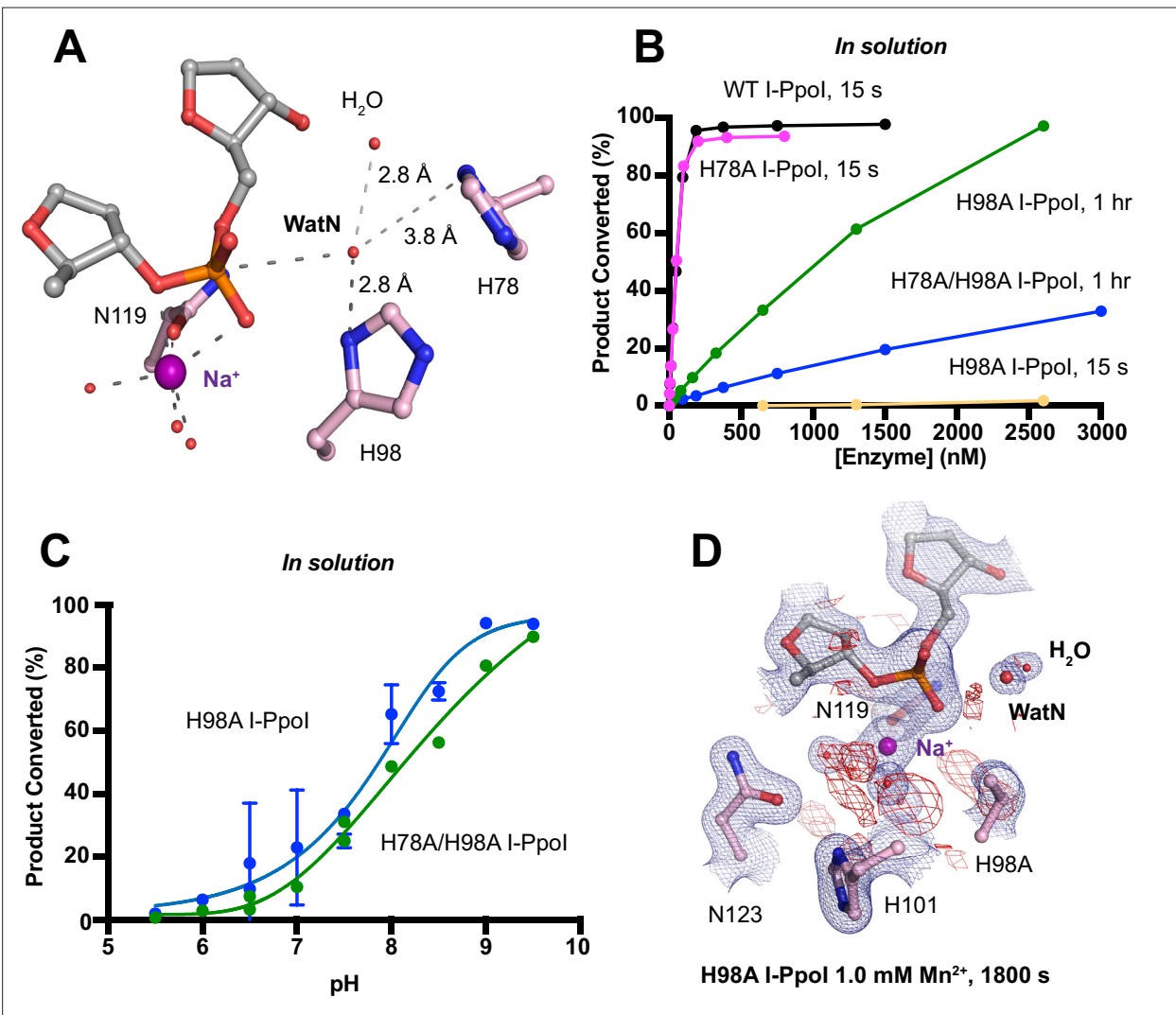

**Figure 5.** Nucleophilic water deprotonation during I-PpoI DNA hydrolysis. (**A**) Active site environment surrounding WatN. His 78 exists near the WatN besides His98. (**B**) In solution DNA hydrolysis activity of various I-PpoI histidine mutants. The points represent the mean of triplicate measurements for the percentage of cleaved reaction product while the error bars (too small to see) represent the standard deviation. (**C**) DNA hydrolysis by H98A I-PpoI (blue) and H78A/H98A I-PpoI (green) at various pH in solution. (**D**) Structure of H98A I-PpoI active site after 1 mM $Mn^{2+}$ soaking for 1800s. The anomalous map for $Mn^{2+}$ was determined at X-ray wavelength of 0.9765 Å and contoured at 2.0 σ. The $2F_o − F_c$ map for $Me^{2+}$, DNA, waters (red spheres), and catalytic residues (blue) was contoured at 2.0 σ. (**B, C**) The points represent the mean of triplicate measurements for the percentage of generated reaction product while the error bars represent the standard deviation.

The online version of this article includes the following figure supplement(s) for figure 5:

**Figure supplement 1.** Histidine I-PpoI mutants and partial rescued cleavage activity by imidazole.

structures that contained 30–35% product at pH 6 and 8 were nearly identical (*Figure 4D, E*). The positions of the His98 residue and the WatN were practically superimposable (*Figure 4F*). At all pH, $Mg^{2+}$ binding strongly correlated with product formation ($R^2 > 0.95$), suggesting that low pH reduces the overall reaction rate without altering the reaction pathway (*Figure 3—figure supplement 1D, E*).

## Nucleophilic water deprotonation pathway during I-PpoI DNA cleavage

We next investigated the deprotonation pathway with mutagenesis. Because His98 has been proposed to primarily activate the nucleophilic water, we first mutated His98 to alanine (*Figure 5A*). As expected, cleavage activity of H98A I-PpoI drastically dropped. Our assays showed that H98A I-PpoI displayed residual activity but required a reaction time of 1 hr to be comparable to WT I-PpoI (*Figure 5B*), similar to the H98Q mutant in previous studies (*Eastberg et al., 2007*). Furthermore, varying the pH resulted in a sigmoidal activity curve of I-PpoI pH dependence, corresponding to a pKa of 7.9 (*Figure 5C*). The significantly reduced reaction rate and the shift of pH dependence suggest that His98 plays a key role in pH sensing and water deprotonation. On the other hand, the low but existing activity of H98A I-PpoI suggests the presence of alternative general bases for proton transfer. Another histidine (His78) resides on the nucleophile side 3.8 Å from the WatN, with a cluster of water molecules in between (*Figure 5A*). We hypothesized that His78 may substitute His98 as the proton acceptor. The DNA cleavage assays revealed that the single mutant, H78A I-PpoI, had an activity similar to WT, whereas the double mutant (H78A/H98A I-PpoI) exhibited much lower activity than that of H98A (*Figure 5B*). Furthermore, varying the pH for H78A/H98A I-PpoI resulted in a sigmoidal activity curve that corresponded to a pKa of 8.7 (*Figure 5C*). In consistent with previous speculations, our results confirm the possibility of His78 as an alternative general base (*Eastberg et al., 2007*). The residual activity and pH dependence of H78A/H98A I-PpoI indicated that something else was still activating the nucleophilic water in the absence of any nearby histidine. Furthermore, we found that titrating imidazole in H98A and H78A/H98A I-PpoI partially rescued cleavage activity (*Figure 5—figure supplement 1A, B*), similar as observed in Cas9 and EndA nuclease (*Furuhata and Kato, 2021*; *Moon et al., 2011*). Collectively, our results indicate that His98 is the primary proton acceptor like previous simulation results (*Maghsoud et al., 2023*) but at the same time, I-PpoI can use alternative pathways to activate the nucleophilic WatN.

We next sought to understand the mechanism of His98-promoted hydrolysis from a structural standpoint. In our *in crystallo* soaking experiments, we equilibrated H98A I-PpoI crystals in 500 μM $Mg^{2+}$. The structure looked nearly identical to the $Na^+$ structure and previous H98A I-PpoI structure with $Mg^{2+}$. As shown earlier, the coordination environment of $Na^+$ and $Mg^{2+}$ was quite similar. To confirm divalent metal ion binding, the H98A I-PpoI crystals were soaked for 1800s in 500 μM $Mn^{2+}$, the same concentration used to initiate DNA hydrolysis by WT I-PpoI. However, to our surprise, the metal ion-binding site was devoid of any anomalous signal. Increasing the $Mn^{2+}$ concentration to 1 mM still did not produce anomalous density at the $Me^{2+}$-binding site (*Figure 5D* and *Figure 5—figure supplement 1C*). The results indicate that metal ion binding can be altered by perturbing His98 and possibly water deprotonation, even though the metal ion-binding site and His98 exist 7 Å apart without direct interaction. Although we tried soaking the H98A I-PpoI crystals in 1 mM Mg/$Mn^{2+}$ and 100 mM imidazole for 15 h, metal ion binding, imidazole binding, or product formation were not detected (*Figure 5—figure supplement 1D*), possibly due to the difficulty of imidazole diffusion within the lattice. Our results indicate that perturbing the deprotonation pathway not only affects nucleophilic attack but also metal binding, suggesting that I-PpoI catalyzes DNA catalysis via a concerted mechanism.

## Discussion

Time-resolved crystallography can visualize time-dependent structural changes and elucidate mechanisms of enzyme catalysis with unparalleled detail *in crystallo*, especially for light-dependent enzymes, in which the reactions can be synchronously initiated by light pulses (*Wilson, 2022*; *Brändén and Neutze, 2021*; *Maestre-Reyna et al., 2023*; *Christou et al., 2023*). In complementary, recently advanced metal ion diffusion-based time-resolved crystallographic techniques have uncovered rich dynamics at the active site and transient metal ion binding during the catalytic processes of metal-ion-dependent DNA polymerases (*Freudenthal et al., 2013*; *Gao and Yang, 2016*; *Nakamura et al.,*

*2012*; *Vyas et al., 2015*; *Chim et al., 2021*; *Gregory et al., 2021*), nucleases (*Samara and Yang, 2018*; *Wu et al., 2019*), and glycosylase (*Demir et al., 2023*). In addition to metal ions captured in static structures, these transient metal ions have been shown to play critical roles in catalysis, such as water deprotonation for nucleophilic attack in RNaseH (*Samara and Yang, 2018*), bond breakage and product stabilization in polymerases (*Freudenthal et al., 2013*; *Gao and Yang, 2016*; *Nakamura et al., 2012*; *Vyas et al., 2015*; *Gregory et al., 2021*; *Reed and Suo, 2017*; *Chang et al., 2022*), and alignment of the substrate and nucleophilic water in MutT (*Nakamura and Yamagata, 2022*). Interestingly, one and only one metal ion was captured within the I-PpoI active site during catalysis, even when high concentration (200 mM $Mn^{2+}$) of metal ion was tested (*Figure 3*). The observed location of the $Me^{2+}$ on the leaving group side of the scissile phosphate corresponds to the $Me^{2+}_B$ in two-metal-ion-dependent nucleases (*Yang, 2008*; *Figure 1—figure supplement 2*). However, the metal ion is unique in its environment and role. First, it is coordinated by a water cluster and asparagine side chain (sometimes with an additional aspartate residue) rather than the acidic aspartate and glutamic acid clusters that outline the active sites of RNaseH and APE1 nuclease (*Tsutakawa et al., 2013*) as well as DNA polymerases (*Figure 6—figure supplement 1*). Even in the absence of divalent $Mg^{2+}$ or $Mn^{2+}$, the active site including the scissile phosphate was already well aligned in I-PpoI, which is again different from RNaseH. Instead, Leu116 and the beta sheet consisting of Arg61, Gln63, Lys65, and Thr67 (*Flick et al., 1998*; *Galburt et al., 2000*) helped to position the DNA optimally toward the active site (*Figure 1—figure supplement 1A, C*). Second, single metal ion binding is strictly correlated with product formation in all conditions, at different pH and with different mutants (*Figure 3A* and *Figure 3—figure supplement 1A–E*; *Moon et al., 2011*). Thus, similar to the third metal ion in DNA polymerases and RNaseH, the metal ion in I-PpoI is not required for substrate alignment but is essential for catalysis. We suspect that the single metal ion helps stabilize the transition state and reduce the electronegative buildup of DNA, thereby promoting DNA hydrolysis.

Proton transfer by a general base is essential for a $SN_2$-type nucleophilic attack. Such deprotonation of the nucleophilic water has been attributed to His98, which is highly conserved in His-Me nucleases. Existing close to the nucleophilic water at 2.6 Å, His98 is perfectly positioned to mediate the proton transfer. Moreover, due to the bulky presence of His98 and beta sheet protein residues, there is no space for an additional metal ion at the nucleophilic side (*Figure 3—figure supplement*

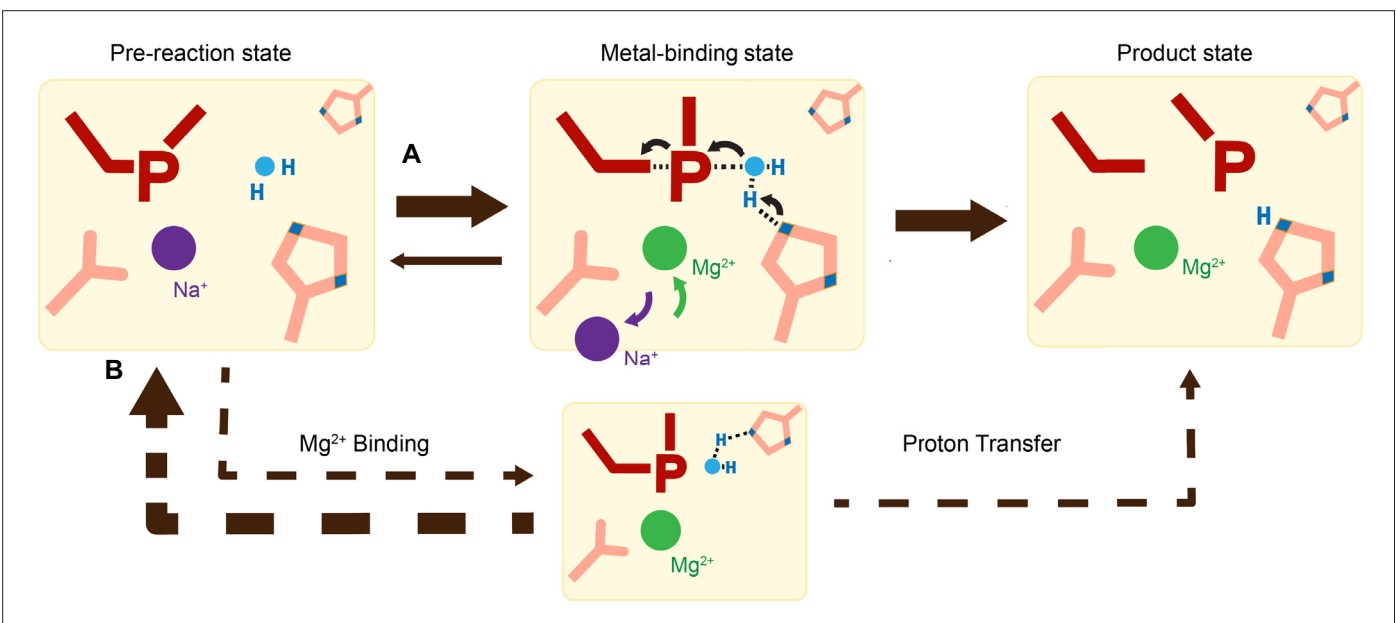

**Figure 6.** Catalytic model of histidine-metal (His-Me) nuclease DNA hydrolysis. Proposed model of His-Me nuclease DNA hydrolysis in which $Me^{2+}$ binding, proton transfer, and nucleophilic attack are concerted (solid arrow) in the presence of the primary proton acceptor in (**A**) versus unfavored (dashed arrows) in the absence of the primary proton acceptor in (**B**).

The online version of this article includes the following figure supplement(s) for figure 6:

**Figure supplement 1.** Active sites of $Me^{2+}$-binding enzymes.

*2*). The H98A mutation significantly reduced catalytic activity and altered pH dependence. But since H98A I-PpoI showed residual activity, His78, imidazole, or its surrounding waters may still serve as alternative general bases for accepting the proton, similar to Pol $\eta$, in which primer 3'-OH deprotonation can occur through multiple pathways (*Gregory et al., 2021*). However, the order of events regarding metal ion binding, water deprotonation, and nucleophilic attack remains unanswered. Based on our *in crystallo* observations, water deprotonation and metal ion binding appeared to be highly correlated (*Figure 6A*). Lowering the pH not only reduced reaction rate but also slowed metal ion binding (*Figure 4B, C*). Moreover, the metal ion was not observed *in crystallo* when His98 was removed (*Figure 5D*). As there is no direct interaction between His98 and the $Me^{2+}$-binding site, the divalent metal ion may be sensitive to the charge potential of the substrate scissile phosphate, which may be indirectly affected by the deprotonated state of the nucleophilic water. Conversely, binding of the divalent metal ion may alter the local electrostatic environment and affect His98 deprotonation. Consistently, previous molecular dynamics simulation of Cas9 has suggested that the histidine pKa is highly sensitive to active site changes (*Nierzwicki et al., 2022*). Without a proper proton acceptor, the metal ion may be prone for dissociation without the reaction proceeding, and thus stable $Mg^{2+}$ binding was not observed *in crystallo* without His98 (*Figure 6B*). On the other hand, optimal alignment of the metal ion and WatN within the active site, labeled as metal-binding state, leads to irreversible bond breakage (*Figure 6A*). In summary, our experimental observations suggest a concerted mechanism for one-metal-ion promoted DNA hydrolysis, offering guidance for future computational analysis of enzyme catalysis (*Nierzwicki et al., 2022*) and the rational design and engineering of nucleases (*Ashworth et al., 2006*) for biotechnological and biomedical applications.

## Materials and methods
### Protein expression and purification
WT, His78Ala, His98Ala, and H78A/H98A *physarum polycephalum* I-PpoI (residues 1–162) were cloned into a modified pET28p vector with a N-terminal 6-histidine tag and a PreScission Protease cleavage site. For protein expression, this I-PpoI plasmid was transformed into BL21 DE3 *E. coli* cells, which were grown in a buffer that contained (10 g/l glucose, 40 g/l α-lactose, 10% glycerol) for 24 hr (20°C). The cell paste was collected via centrifugation and re-suspended in a buffer that contained 50 mM Tris (pH 7.5), 1 M NaCl, 1 mM $MgCl_2$, 10 mM imidazole, 2 mM β-mercaptoethanol (BME), and 5% glycerol. After sonication, I-PpoI was loaded onto a HisTrap HP column (GE Healthcare), which was pre-equilibrated with a buffer that contained 50 mM Tris (pH 7.5), 1 M NaCl, 1 mM $MgCl_2$, 10 mM imidazole, 2 mM BME, and 5% glycerol. The column was washed with 300 ml of buffer to remove non-specific bound proteins and was eluted with buffer that contained 50 mM Tris (pH 7.5), 1 M NaCl, 1 mM $MgCl_2$, 300 mM imidazole, and 2 mM BME. The eluted I-PpoI was incubated with PreScission Protease to cleave the N-terminal 6-histidine-tag. Afterwards, I-PpoI was desalted to 50 mM Tris (pH 7.5), 167 mM NaCl, 1 mM $MgCl_2$, 2 mM BME, and 5% glycerol and was loaded onto a Heparin column (GE Healthcare) equilibrated with 50 mM Tris (pH 7.5) and 167 mM NaCl. The protein was eluted with an increasing salt (NaCl) gradient, concentrated, and stored at 40% glycerol at −80°C.

### DNA hydrolysis assays
DNA hydrolysis activity of varying time was assayed by the following: The reaction mixture contained 100 nM WT I-PpoI, 50 mM NaCl, 100 mM Tris (pH7.5), 1.5 mM dithiothreitol (DTT), 0.05 mg/ml bovine serum albumin (BSA), 50 nM DNA, 10 µM ethylenediaminetetraacetic acid (EDTA), and 4% glycerol. The hydrolysis assays were executed using a palindromic 5'-fluorescein-labeled DNA duplex (5'-TTG ACT CTC TTA AGA GAG TCA-3'). Reactions were initiated by adding 10 mM $MgCl_2$ to the reaction mixture for 0–1 hr at 37°C and were stopped by mixing with equal volume of a quench buffer, which contain 80% formamide, 100 mM EDTA (pH 8.0), 0.2 mg/ml xylene cyanol, and 0.2 mg/ml bromophenol.

 The DNA hydrolysis activity at different pH was assayed by the following: The reaction mixture contained 100–3000 nM WT, H98A, and H78A/H98A I-PpoI, 50 mM NaCl, 1.5 mM DTT, 0.05 mg/ml BSA, 50 nM DNA, 10 µM EDTA, and 4% glycerol. Reactions were initiated by adding 50 mM 2-(N-morpholino)ethanesulfonic acid (MES) (pH 5.5–6.5), 50 mM 4-(2-hydroxyethyl)-1-piperazineetha

nesulfonic acid (HEPES) (pH 6.5–7.5), or 50 mM Tris (pH 7.5–9.5) together with 10 mM $MgCl_2$ to the reaction mixture for 30 min at 37°C and were stopped by adding equal volume of quench buffer.

The DNA hydrolysis activity with different mutants was assayed by the following: The reaction mixture contained 50 mM NaCl, 100 mM Tris (pH7.5), 1.5 mM DTT, 0.05 mg/ml BSA, 50 nM DNA, 10 mM $MgCl_2$, 10 μM EDTA, and 4% glycerol. Reactions were initiated by adding 0–3000 nM of WT, H78A, H98A, H78A/H98A I-PpoI to the reaction mixture for 15 s or 1 hr at 37°C and stopped by adding equal volume of quench buffer.

The DNA hydrolysis activity with different metal ions was assayed by the following: The reaction mixture contained 100 nM WT I-PpoI, 50 mM NaCl, 100 mM Tris (pH 7.5), 1.5 mM DTT, 0.05 mg/ml BSA, 50 nM DNA, 10 μM EDTA, and 4% glycerol. Reactions were initiated by adding 10 mM of $MgCl_2$, $MnCl_2$, $CaCl_2$, $NiCl_2$, and $ZnCl_2$ to the reaction mixture for 15 s at 37°C and were stopped by adding equal volume of quench buffer.

The DNA hydrolysis activity with $Tl^+$ or additional $Na^+$ was assayed by the following: The reaction mixture contained 100–3000 nM WT I-PpoI, 100 mM Tris (pH7.5), 1.5 mM DTT, 0.05 mg/ml BSA, 50 nM DNA, 10 μM EDTA, and 4% glycerol. Reactions were conducted at 37°C for 30 min by adding 0–350 mM TlCl or NaCl together with 10 mM $MgCl_2$ to the reaction mixture and were stopped by adding equal volume of quench buffer.

The DNA hydrolysis activity with imidazole was assayed by the following: The reaction mixture contained 100–3000 nM H98A and H78A/H98A I-PpoI, 50 mM NaCl, 100 mM Tris (pH7.5), 1.5 mM DTT, 0.05 mg/ml BSA, 50 nM DNA, 10 μM EDTA, and 4% glycerol. Reactions were conducted at 37°C for 30 min by adding 0–100 mM imidazole together with 10 mM $MgCl_2$ to the reaction mixture and were stopped by adding equal volume of quench buffer.

For all reactions, after heating the quenched reaction mix to 97°C for 5 min and immediately placing on ice, reaction products were resolved on 22.5% polyacrylamide urea gels. The gels were visualized by a Sapphire Biomolecular Imager and quantified using the built-in software. Quantification of percentage cleaved and graphic representation were executed by Graph Prism.

## Crystallization

WT or H98A I-PpoI in a buffer containing 20 mM Tris 7.5, 300 mM NaCl, 3 mM DTT, and 0.1 mM EDTA was added with (5′-TTG ACT CTC TTA AGA GAG TCA-3′) DNA at a molar molar of 1:1.5 for I-PpoI and DNA and added with threefolds volume of buffer that contained 20 mM Tris 7.5, 3 mM DTT, and 0.1 mM EDTA. This I-PpoI–DNA complex was then cleaned with a Superdex 200 10/300 GL column (GE Healthcare) with a buffer that contained 20 mM Tris 7.5, 150 mM NaCl, 3 mM DTT, and 0.1 mM EDTA. The I-PpoI–DNA complex was concentrated to 2.8 mg/ml I-PpoI (confirmed by Bradford assay). All crystals were obtained using the hanging-drop vapor-diffusion method against a reservoir solution containing 0.1 M MES (pH 6.0), 0.2 M sodium malonate, and 20% (wt/vol) PEG3350 at room temperature within 4 days.

To identify the monovalent $Me^+$ species that binds during the pre-reaction state, WT I-PpoI crystals were transferred and incubated in a buffer containing 0.1 M MES (pH 6.0), 70 mM thallium acetate and 20% (wt/vol) PEG3350 for 30 min. Afterwards, the crystals were quickly dipped in a cryo-solution supplemented with 20% (wt/vol) glycerol and flash-cooled in liquid nitrogen.

## Chemical reaction *in crystallo*

The WT I-PpoI crystals were first transferred and incubated in a pre-reaction buffer containing 0.1 M MES (pH 6.0 or 7.0) or 0.1 M Tris (pH 8.0), 0.2 M NaCl, and 20% (wt/vol) PEG3350 for 30 min. The chemical reaction was initiated by transferring the crystals into a reaction buffer containing 0.1 M MES (pH 6.0 or 7.0) or 0.1 M Tris (pH 8.0), 0.2 M NaCl, and 20% (wt/vol) PEG3350, and 500 μM $MgCl_2$ or $MnCl_2$. After incubation for a desired time period, the crystals were quickly dipped in a cryo-solution supplemented with 20% (wt/vol) glycerol and flash-cooled in liquid nitrogen.

To observe any additional $Me^{2+}$ binding sites during DNA hydrolysis, WT I-PpoI crystals were first transferred and incubated in a pre-reaction buffer containing 0.1 M MES (pH 6.0), 0.2 M NaCl, and 20% (wt/vol) PEG3350 for 30 min. The chemical reaction was initiated by transferring the crystals into a reaction buffer containing 0.1 M MES (pH 6.0), 0.2 M NaCl, and 20% (wt/vol) PEG3350, and 200 mM $MnCl_2$. After incubation for 600 s, the crystals were quickly dipped in a cryo-solution supplemented with 20% (wt/vol) glycerol and flash-cooled in liquid nitrogen.

The metal ion soaking experiments with His98Ala I-PpoI were performed following the similar protocol as that of WT I-PpoI. His98Ala I-PpoI crystals were first incubated in a pre-reaction buffer containing 0.1 M MES 7.0, 0.2 M NaCl, and 20% (wt/vol) PEG3350 for 30 min, followed by 1800s incubation in a reaction buffer containing 0.1 M MES 7.0, 0.2 M NaCl, and 20% (wt/vol) PEG3350, and 1 mM $MnCl_2$. To observe whether soaking in imidazole can initiate the reaction in the absence of His98, His98Ala I-PpoI crystals were first transferred and incubated in a buffer containing 0.1 M MES (pH 6.0), 0.2 M NaCl, 1 mM $MnCl_2$, and 20% (wt/vol) PEG3350 for 30 min. The crystals were then transferred into a reaction buffer containing 0.1 M MES (pH 6.0), 0.2 M NaCl, 1 mM $MgCl_2$ or $MnCl_2$, and 100 mM imidazole, and 20% (wt/vol) PEG3350. After incubation for a desired time period, the crystals were quickly dipped in a cryo-solution supplemented with 20% (wt/vol) glycerol and flash-cooled in liquid nitrogen.

## Data collection and refinement

Diffraction data were collected at 100 K on LS-CAT beam lines 21-D-D, 21-ID-F, and 21-ID-G at 1.1 or 0.97 Å at the Advanced Photon Source (Argonne National Laboratory) or beamlines 5.0.3 at 0.97 Å at Advanced Light Source (ALS). Data were indexed in space group P3$_1$21, scaled with XSCALE and reduced using X-ray Detector Software (XDS) (**Kabsch, 2010**). Isomorphous I-PpoI structures with Na$^+$ PDB ID 1CZ0 were used as initial models for refinement using PHENIX (**Adams et al., 2010**) and COOT (**Emsley et al., 2010**).

Occupancies were assigned for the reaction product until there were no significant $F_o - F_c$ peaks. Occupancies were assigned for the metal ions, following the previous protocol (**Gao and Yang, 2016**) until (1) there were no significant $F_o - F_c$ peaks, (2) the B value had roughly similar values to its ligand (3) it matched the occupancy of the reaction product. For the structures in which some $F_o - F_c$ peaks were present around the Me$^{2+}$ binding sites or reaction product, no change in the assigned occupancy was executed when a 10% change in occupancy (e.g. 100–90%) failed to significantly change the intensity of the $F_o - F_c$ peaks. Source data of the electron densities in r.m.s. density are provided as a Source Data file. Each structure was refined to the highest resolution data collected, which ranged between 1.42 and 2.2 Å. Software applications used in this project were compiled and configured by SBGrid (**Morin et al., 2013**). Source data of data collection and refinement statistics are summarized in **Supplementary file 1**. All structural figures were drawn using PyMOL (https://www.pymol.org/).

## Calculation of electron density

Electron density (r.m.s.d.) from the $F_o - F_c$ map of the product phosphate and Me$^{2+}$ were calculated by running a round of B-factor refinement in PHENIX after omitting the reaction product phosphate and Me$^{2+}$ atoms from phase calculation in COOT. All structures from the same experiment (Mg$^{2+}$-pH 6.0, Mg$^{2+}$-pH 7.0, Mg$^{2+}$-pH 8.0, Mn$^{2+}$-pH 6.0) were refined to the lowest resolution in the same experiment group (1.80 Å for Mg$^{2+}$-pH 6.0, 1.59 Å for Mg$^{2+}$-pH 7.0, 1.70 Å for Mg$^{2+}$-pH 8.0, and 1.80 Å for Mn$^{2+}$-pH 6.0).

## Acknowledgements

Our sincere appreciation to the members of the Gao lab, and Drs. Phillips, Nikonowicz, and Lu who serve on CC's thesis committee. We thank the APS LS-CAT beam technicians and research scientists Drs. Anderson, Wawrzak, Brunzelle, and Focia. This research used resources of the Advanced Photon Source, a U.S. Department of Energy (DOE) Office of Science User Facility operated for the DOE Office of Science by Argonne National Laboratory under Contract No. DE-AC02-06CH11357. Use of the LS-CAT Sector 21 was supported by the Michigan Economic Development Corporation and the Michigan Technology Tri-Corridor (Grant 085P1000817). The Berkeley Center for Structural Biology is supported in part by the Howard Hughes Medical Institute. The Advanced Light Source is a Department of Energy Office of Science User Facility under Contract No. DE-AC02-05CH11231. The Pilatus detector on 5.0.1 was funded under NIH grant S10OD021832. The ALS-ENABLE beamlines are supported in part by the National Institutes of Health, National Institute of General Medical Sciences, grant P30 GM124169. This work is supported by CPRIT (RR190046) and the Welch Foundation (C-2033-20200401) to YG, a predoctoral fellowship from the Houston Area Molecular Biophysics Program (NIH Grant No. T32 GM008280, Program Director Dr. Theodore Wensel) to CC.

## Additional information

### Funding

| Funder | Grant reference number | Author |
|---|---|---|
| Cancer Prevention and Research Institute of Texas | RR190046 | Yang Gao |
| Welch Foundation | C-2033- 624 20200401 | Yang Gao |
| National Institutes of Health | T32 GM008280 | Caleb Chang |

The funders had no role in study design, data collection, and interpretation, or the decision to submit the work for publication.

### Author contributions

Caleb Chang, Conceptualization, Data curation, Formal analysis, Funding acquisition, Investigation, Writing - original draft, Writing - review and editing; Grace Zhou, Data curation, Writing - original draft; Yang Gao, Conceptualization, Funding acquisition, Writing - original draft, Writing - review and editing

### Author ORCIDs

Caleb Chang (ID) http://orcid.org/0000-0002-4507-659X
Grace Zhou (ID) https://orcid.org/0009-0009-9021-8305
Yang Gao (ID) https://orcid.org/0000-0002-4037-0431

Reviewer #1 (Public Review): https://doi.org/10.7554/eLife.99960.3.sa1
Reviewer #2 (Public Review): https://doi.org/10.7554/eLife.99960.3.sa2
Author response https://doi.org/10.7554/eLife.99960.3.sa3

## Additional files

### Supplementary files
- Supplementary file 1. Crystal diffraction and refinement data.
- MDAR checklist

### Data availability

The coordinates, density maps, and structure factors for all the structures have been deposited in Protein Data Bank (PDB) under accession codes: 8VMO, 8VMP, 8VMQ, 8VMR, 8VMS, 8VMT, 8VMU, 8VMV, 8VMW, 8VMX, 8VMY, 8VMZ, 8VN0, 8VN1, 8VN2, 8VN3, 8VN4, 8VN5, 8VN6, 8VN7, 8VN8, 8VN9, 8VNA, 8VNB, 8VNC, 8VND, 8VNE, 8VNF, 8VNG, 8VNH, 8VNJ, 8VNK, 8VNL, 8VNM, 8VNN, 8VNO, 8VNP, 8VNQ, 8VNR, 8VNS, 8VNT, and 8VNU. All data are available in the main text or the supplementary materials.

The following datasets were generated:

| Author(s) | Year | Dataset title | Dataset URL | Database and Identifier |
|---|---|---|---|---|
| Chang C, Gao Y | 2024 | Homing endonuclease I-PpoI-DNA complex:ground state at pH7.0 (K+ MES) with Na+ | https://www.rcsb.org/structure/8VMO | RCSB Protein Data Bank, 8VMO |
| Chang C, Gao Y | 2024 | Homing endonuclease I-PpoI-DNA complex:reaction at pH7.0 (K+ MES) with 500 uM Mg2+ for 10s | https://www.rcsb.org/structure/8VMP | RCSB Protein Data Bank, 8VMP |

*Continued on next page*

*Continued*

| Author(s) | Year | Dataset title | Dataset URL | Database and Identifier |
|---|---|---|---|---|
| Chang C, Gao Y | 2024 | Homing endonuclease I-PpoI-DNA complex:reaction at pH7.0 (K+ MES) with 500 uM Mg2+ for 20s | https://www.rcsb.org/structure/8VMQ | RCSB Protein Data Bank, 8VMQ |
| Chang C, Gao Y | 2024 | Homing endonuclease I-PpoI-DNA complex:reaction at pH7.0 (K+ MES) with 500 uM Mg2+ for 40s | https://www.rcsb.org/structure/8VMR | RCSB Protein Data Bank, 8VMR |
| Chang C, Gao Y | 2024 | Homing endonuclease I-PpoI-DNA complex:reaction at pH7.0 (K+ MES) with 500 uM Mg2+ for 80s | https://www.rcsb.org/structure/8VMS | RCSB Protein Data Bank, 8VMS |
| Chang C, Gao Y | 2024 | Homing endonuclease I-PpoI-DNA complex:reaction at pH7.0 (K+ MES) with 500 uM Mg2+ for 160s | https://www.rcsb.org/structure/8VMT | RCSB Protein Data Bank, 8VMT |
| Chang C, Gao Y | 2024 | Homing endonuclease I-PpoI-DNA complex:reaction at pH7.0 (K+ MES) with 500 uM Mg2+ for 320s | https://www.rcsb.org/structure/8VMU | RCSB Protein Data Bank, 8VMU |
| Chang C, Gao Y | 2024 | Homing endonuclease I-PpoI-DNA complex:reaction at pH7.0 (K+ MES) with 500 uM Mg2+ for 600s | https://www.rcsb.org/structure/8VMV | RCSB Protein Data Bank, 8VMV |
| Chang C, Gao Y | 2024 | Homing endonuclease I-PpoI-DNA complex:ground state at pH6.0 (K+ MES) with Na+ | https://www.rcsb.org/structure/8VMW | RCSB Protein Data Bank, 8VMW |
| Chang C, Gao Y | 2024 | Homing endonuclease I-PpoI-DNA complex:reaction at pH6.0 (K+ MES) with 500 uM Mg2+ for 10s | https://www.rcsb.org/structure/8VMX | RCSB Protein Data Bank, 8VMX |
| Chang C, Gao Y | 2024 | Homing endonuclease I-PpoI-DNA complex:reaction at pH6.0 (K+ MES) with 500 uM Mg2+ for 20s | https://www.rcsb.org/structure/8VMY | RCSB Protein Data Bank, 8VMY |
| Chang C, Gao Y | 2024 | Homing endonuclease I-PpoI-DNA complex:reaction at pH6.0 (K+ MES) with 500 uM Mg2+ for 40s | https://www.rcsb.org/structure/8VMZ | RCSB Protein Data Bank, 8VMZ |
| Chang C, Gao Y | 2024 | Homing endonuclease I-PpoI-DNA complex:reaction at pH6.0 (K+ MES) with 500 uM Mg2+ for 80s | https://www.rcsb.org/structure/8VN0 | RCSB Protein Data Bank, 8VN0 |
| Chang C, Gao Y | 2024 | Homing endonuclease I-PpoI-DNA complex:reaction at pH6.0 (K+ MES) with 500 uM Mg2+ for 160s | https://www.rcsb.org/structure/8VN1 | RCSB Protein Data Bank, 8VN1 |

*Continued*

| Author(s) | Year | Dataset title | Dataset URL | Database and Identifier |
|---|---|---|---|---|
| Chang C, Gao Y | 2024 | Homing endonuclease I-PpoI-DNA complex:reaction at pH6.0 (K+ MES) with 500 uM Mg2+ for 320s | https://www.rcsb.org/structure/8VN2 | RCSB Protein Data Bank, 8VN2 |
| Chang C, Gao Y | 2024 | Homing endonuclease I-PpoI-DNA complex:reaction at pH6.0 (K+ MES) with 500 uM Mg2+ for 600s | https://www.rcsb.org/structure/8VN3 | RCSB Protein Data Bank, 8VN3 |
| Chang C, Gao Y | 2024 | Homing endonuclease I-PpoI-DNA complex:reaction at pH6.0 (K+ MES) with 500 uM Mg2+ for 1200s | https://www.rcsb.org/structure/8VN4 | RCSB Protein Data Bank, 8VN4 |
| Chang C, Gao Y | 2024 | Homing endonuclease I-PpoI-DNA complex:ground state at pH8.0 (Tris) with Na+ | https://www.rcsb.org/structure/8VN5 | RCSB Protein Data Bank, 8VN5 |
| Chang C, Gao Y | 2024 | Homing endonuclease I-PpoI-DNA complex:reaction at pH8.0 (Tris) with 500 uM Mg2+ for 10s | https://www.rcsb.org/structure/8VN6 | RCSB Protein Data Bank, 8VN6 |
| Chang C, Gao Y | 2024 | Homing endonuclease I-PpoI-DNA complex:reaction at pH8.0 (Tris) with 500 uM Mg2+ for 20s | https://www.rcsb.org/structure/8VN7 | RCSB Protein Data Bank, 8VN7 |
| Chang C, Gao Y | 2024 | Homing endonuclease I-PpoI-DNA complex:reaction at pH8.0 (Tris) with 500 uM Mg2+ for 40s | https://www.rcsb.org/structure/8VN8 | RCSB Protein Data Bank, 8VN8 |
| Chang C, Gao Y | 2024 | Homing endonuclease I-PpoI-DNA complex:reaction at pH8.0 (Tris) with 500 uM Mg2+ for 80s | https://www.rcsb.org/structure/8VN9 | RCSB Protein Data Bank, 8VN9 |
| Chang C, Gao Y | 2024 | Homing endonuclease I-PpoI-DNA complex:reaction at pH8.0 (Tris) with 500 uM Mg2+ for 160s | https://www.rcsb.org/structure/8VNA | RCSB Protein Data Bank, 8VNA |
| Chang C, Gao Y | 2024 | Homing endonuclease I-PpoI-DNA complex:reaction at pH8.0 (Tris) with 500 uM Mg2+ for 240s | https://www.rcsb.org/structure/8VNB | RCSB Protein Data Bank, 8VNB |
| Chang C, Gao Y | 2024 | Homing endonuclease I-PpoI-DNA complex:reaction at pH8.0 (Tris) with 500 uM Mg2+ for 320s | https://www.rcsb.org/structure/8VNC | RCSB Protein Data Bank, 8VNC |
| Chang C, Gao Y | 2024 | Homing endonuclease I-PpoI-DNA complex:reaction at pH8.0 (Tris) with 500 uM Mg2+ for 600s | https://www.rcsb.org/structure/8VND | RCSB Protein Data Bank, 8VND |

*Continued on next page*

*Continued*

| Author(s) | Year | Dataset title | Dataset URL | Database and Identifier |
|---|---|---|---|---|
| Chang C, Gao Y | 2024 | Homing endonuclease I-PpoI-DNA complex:reaction at pH6.0 (K+ MES) with 500 uM Mn2+ for 10s | https://www.rcsb.org/structure/8VNE | RCSB Protein Data Bank, 8VNE |
| Chang C, Gao Y | 2024 | Homing endonuclease I-PpoI-DNA complex:reaction at pH6.0 (K+ MES) with 500 uM Mn2+ for 20s | https://www.rcsb.org/structure/8VNF | RCSB Protein Data Bank, 8VNF |
| Chang C, Gao Y | 2024 | Homing endonuclease I-PpoI-DNA complex:reaction at pH6.0 (K+ MES) with 500 uM Mn2+ for 40s | https://www.rcsb.org/structure/8VNG | RCSB Protein Data Bank, 8VNG |
| Chang C, Gao Y | 2024 | Homing endonuclease I-PpoI-DNA complex:reaction at pH6.0 (K+ MES) with 500 uM Mn2+ for 80s | https://www.rcsb.org/structure/8VNH | RCSB Protein Data Bank, 8VNH |
| Chang C, Gao Y | 2024 | Homing endonuclease I-PpoI-DNA complex:reaction at pH6.0 (K+ MES) with 500 uM Mn2+ for 120s | https://www.rcsb.org/structure/8VNJ | RCSB Protein Data Bank, 8VNJ |
| Chang C, Gao Y | 2024 | Homing endonuclease I-PpoI-DNA complex:reaction at pH6.0 (K+ MES) with 500 uM Mn2+ for 160s | https://www.rcsb.org/structure/8VNK | RCSB Protein Data Bank, 8VNK |
| Chang C, Gao Y | 2024 | Homing endonuclease I-PpoI-DNA complex:reaction at pH6.0 (K+ MES) with 500 uM Mn2+ for 240s | https://www.rcsb.org/structure/8VNL | RCSB Protein Data Bank, 8VNL |
| Chang C, Gao Y | 2024 | Homing endonuclease I-PpoI-DNA complex:reaction at pH6.0 (K+ MES) with 500 uM Mn2+ for 320s | https://www.rcsb.org/structure/8VNM | RCSB Protein Data Bank, 8VNM |
| Chang C, Gao Y | 2024 | Homing endonuclease I-PpoI-DNA complex:reaction at pH6.0 (K+ MES) with 500 uM Mn2+ for 480s | https://www.rcsb.org/structure/8VNN | RCSB Protein Data Bank, 8VNN |
| Chang C, Gao Y | 2024 | Homing endonuclease I-PpoI-DNA complex:reaction at pH6.0 (K+ MES) with 500 uM Mn2+ for 600s | https://www.rcsb.org/structure/8VNO | RCSB Protein Data Bank, 8VNO |
| Chang C, Gao Y | 2024 | Homing endonuclease I-PpoI-DNA complex at pH6.0 (K+ MES) with 0.2 M sodium malonate | https://www.rcsb.org/structure/8VNP | RCSB Protein Data Bank, 8VNP |
| Chang C, Gao Y | 2024 | Homing endonuclease H98A I-PpoI-DNA complex at pH6.0 (K+ MES) with 1 mM Mn2+ for 1800s | https://www.rcsb.org/structure/8VNQ | RCSB Protein Data Bank, 8VNQ |

*Continued on next page*

*Continued*

| Author(s) | Year | Dataset title | Dataset URL | Database and Identifier |
|---|---|---|---|---|
| Chang C, Gao Y | 2024 | Homing endonuclease H98A I-PpoI-DNA complex at pH6.0 (K+ MES) with 1 mM Mn2+ for 600s and then 100 mM imidazole for 15 hrs | https://www.rcsb.org/structure/8VNR | RCSB Protein Data Bank, 8VNR |
| Chang C, Gao Y | 2024 | Homing endonuclease I-PpoI-DNA complex:reaction at pH6.0 (K+ MES) with 200 mM Mn2+ for 600s | https://www.rcsb.org/structure/8VNS | RCSB Protein Data Bank, 8VNS |
| Chang C, Gao Y | 2024 | Homing endonuclease I-PpoI-DNA complex:reaction at pH6.0 (K+ MES) with 500 uM Mg2+ for 1800s | https://www.rcsb.org/structure/8VNT | RCSB Protein Data Bank, 8VNT |
| Chang C, Gao Y | 2024 | Homing endonuclease H98A I-PpoI-DNA complex at pH6.0 (K+ MES) with 70 mM Tl+ for 1800s | https://www.rcsb.org/structure/8VNU | RCSB Protein Data Bank, 8VNU |

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
