## [Editor Report · eLife assessment]

Chang et al. have investigated the catalytic mechanism of I-PpoI nuclease, a one-metal-ion dependent nuclease, by time-resolved X-ray crystallography using soaking of crystals with metal ions under different pH conditions. This **convincing** study revealed that I-PpoI catalyzes the reaction process through a single divalent cation. The study uncovers **important** details of the roles of the metal ion and the active site histidine in catalysis.

---

## [Referee Report · Reviewer #1 (Public Review)]

This study is convincing because they performed time-resolved X-ray crystallography under different pH conditions using active/inactive metal ions and PpoI mutants, as with the activity measurements in solution in conventional enzymatic studies. Although the reaction mechanism is simple and maybe a little predictable, the strength of this study is that they were able to validate that PpoI catalyzes DNA hydrolysis through "a single divalent cation" because time-resolved X-ray study often observes transient metal ions which are important for catalysis but are not predictable in previous studies with static structures such as enzyme-substrate analog-metal ion complexes. The discussion of this study is well supported by their data. This study visualized the catalytic process and mutational effects on catalysis, providing a new insight into the catalytic mechanism of I-PpoI through a single divalent cation. The authors found that His98, a candidate of proton acceptor in the previous experiments, also affects the Mg2+ binding for catalysis without the direct interaction between His98 and the Mg2+ ion, suggesting that "Without a proper proton acceptor, the metal ion may be prone for dissociation without the reaction proceeding, and thus stable Mg2+ binding was not observed in crystallo without His98". In the future, this interesting feature observed in I-PpoI should be investigated by biochemical, structural and computational analyses using other one metal-ion dependent nucleases.

---

## [Referee Report · Reviewer #2 (Public Review)]

Summary:

Most polymerases and nucleases use two or three divalent metal ions in their catalytic functions. The family of His-Me nucleases, however, use only one divalent metal ion, along with a conserved histidine, to catalyze DNA hydrolysis. The mechanism has been studied previously but, according to the authors, it remained unclear. By use of time resolved X-ray crystallography, this work convincingly demonstrated that only one M2+ ion is involved in the catalysis of the His-Me I-PpoI 19 nuclease, and proposed concerted functions of the metal and the histidine.

Strengths:

This work performs mechanistic studies, including the number and roles of metal ion, pH dependence, and activation mechanism, all by structural analyses, coupled with some kinetics and mutagenesis. Overall, it is a highly rigorous work. This approach was first developed in Science (2016) for a DNA polymerase, in which Yang Cao was the first author. It has subsequently been applied to just 5 to 10 enzymes by different labs, mainly to clarify two versus three metal ion mechanisms. The present study is the first one to demonstrate a single metal ion mechanism by this approach.

Furthermore, on the basis of the quantitative correlation between the fraction of metal ion binding and the formation of product, as well as the pH dependence, and the data from site specific mutants, the authors concluded that the functions of Mg2+ and His are a concerted process. A detailed mechanism is proposed in Figure 6.

Even though there are no major surprises in the results and conclusions, the time-resolved structural approach and the overall quality of the results represent a significant step forward for the Me-His family of nucleases. In addition, since the mechanism is unique among different classes of nucleases and polymerases, the work should be of interest to readers in DNA enzymology, or even mechanistic enzymology in general.

Weaknesses:

Two relatively minor issues are raised here for consideration by the authors:

p. 4, last para, lines 1-2: "we next visualized the entire reaction process by soaking I-PpoI crystals in buffer....". This is a little over-stated. The structures being observed are not reaction intermediates. They are mixtures of substrates and products in the enzyme-bound state. The progress of the reaction is limited by the progress of soaking of the metal ion. Crystallography is just been used as a tool to monitor the reaction (and provide structural information about the product). It would be more accurate to say that "we next monitored the reaction progress by soaking...."

p. 5, beginning of the section. The authors on one hand emphasized the quantitative correlation between Mg ion density and the product density. On the other hand, they raised the uncertainty in the quantitation of Mg2+ density versus Na+ density, thus they repeated the study with Mn2+ which has distinct anomalous signals. This is a very good approach. However, still no metal ion density is shown in the key figure 2A. It will be clearer to show the progress of metal ion density in a figure (in addition to just plots), whether it is Mg or Mn.

Revised version: The authors have properly revised the paper in response to both questions raised in the weakness section. The first issue is an important clarification for others working on similar approaches also. For the second issue, the metal ion density is nicely shown in Fig. S4 now.

---

## [Author Response]

The following is the authors’ response to the original reviews.

**Public Reviews:**

**Reviewer #1 (Public Review):**
This study is convincing because they performed time-resolved X-ray crystallography under different pH conditions using active/inactive metal ions and PpoI mutants, as with the activity measurements in solution in conventional enzymatic studies. Although the reaction mechanism is simple and may be a little predictable, the strength of this study is that they were able to validate that PpoI catalyzes DNA hydrolysis through "a single divalent cation" because time-resolved X-ray study often observes transient metal ions which are important for catalysis but are not predictable in previous studies with static structures such as enzyme-substrate analog-metal ion complexes. The discussion of this study is well supported by their data. This study visualized the catalytic process and mutational effects on catalysis, providing new insight into the catalytic mechanism of I-PpoI through a single divalent cation. The authors found that His98, a candidate of proton acceptor in the previous experiments, also affects the Mg2+ binding for catalysis without the direct interaction between His98 and the Mg2+ ion, suggesting that "Without a proper proton acceptor, the metal ion may be prone for dissociation without the reaction proceeding, and thus stable Mg2+ binding was not observed in crystallo without His98". In future, this interesting feature observed in I-PpoI should be investigated by biochemical, structural, and computational analyses using other metal-ion dependent nucleases.

We appreciate the reviewer for the positive assessment as well as all the comments and suggestions.

**Reviewer #2 (Public Review):**
Summary:Most polymerases and nucleases use two or three divalent metal ions in their catalytic functions. The family of His-Me nucleases, however, use only one divalent metal ion, along with a conserved histidine, to catalyze DNA hydrolysis. The mechanism has been studied previously but, according to the authors, it remained unclear. By use of a time resolved X-ray crystallography, this work convincingly demonstrated that only one M2+ ion is involved in the catalysis of the His-Me I-PpoI 19 nuclease, and proposed concerted functions of the metal and the histidine.Strengths:This work performs mechanistic studies, including the number and roles of metal ion, pH dependence, and activation mechanism, all by structural analyses, coupled with some kinetics and mutagenesis. Overall, it is a highly rigorous work. This approach was first developed in Science (2016) for a DNA polymerase, in which Yang Cao was the first author. It has subsequently been applied to just 5 to 10 enzymes by different labs, mainly to clarify two versus three metal ion mechanisms. The present study is the first one to demonstrate a single metal ion mechanism by this approach.Furthermore, on the basis of the quantitative correlation between the fraction of metal ion binding and the formation of product, as well as the pH dependence, and the data from site-specific mutants, the authors concluded that the functions of Mg2+ and His are a concerted process. A detailed mechanism is proposed in Figure 6.Even though there are no major surprises in the results and conclusions, the time-resolved structural approach and the overall quality of the results represent a significant step forward for the Me-His family of nucleases. In addition, since the mechanism is unique among different classes of nucleases and polymerases, the work should be of interest to readers in DNA enzymology, or even mechanistic enzymology in general.

Thank you very much for your comments and suggestions.

Weaknesses:Two relatively minor issues are raised here for consideration:p. 4, last para, lines 1-2: "we next visualized the entire reaction process by soaking I-PpoI crystals in buffer....". This is a little over-stated. The structures being observed are not reaction intermediates. They are mixtures of substrates and products in the enzyme-bound state. The progress of the reaction is limited by the progress of the soaking of the metal ion. Crystallography has just been used as a tool to monitor the reaction (and provide structural information about the product). It would be more accurate to say that "we next monitored the reaction progress by soaking....".

We appreciate the clarification regarding the description of our experimental approach. We agree that our structures do not represent reaction intermediates but rather mixtures of substrate and product states within the enzyme-bound environment. We have revised the text accordingly to more accurately reflect our methodology.

p. 5, the beginning of the section. The authors on one hand emphasized the quantitative correlation between Mg ion density and the product density. On the other hand, they raised the uncertainty in the quantitation of Mg2+ density versus Na+ density, thus they repeated the study with Mn2+ which has distinct anomalous signals. This is a very good approach. However, there is still no metal ion density shown in the key Figure 2A. It will be clearer to show the progress of metal ion density in a figure (in addition to just plots), whether it is Mg or Mn.

Thank you for your insightful comments. We recognize the importance of visualizing metal ion density alongside product density data. To address this, we included in Figure S4 to present Mg2+/Mn2+ and product densities concurrently.

**Reviewer #1 (Recommendations For The Authors):**
(1) Figure 6. I understand that pre-reaction state (left panel) and Metal-binding state (two middle panels) are in equilibrium. But can we state that the Metal-binding state (two middle panels) and the product state (right panel) are in equilibrium and connected by two arrows?

Thank you for your comments. We agree that the DNA hydrolysis reaction process may not be reversible within I-Ppo1 active site. To clarify, we removed the backward arrows between the metal-binding state and product state. In addition, we thank the reviewer for giving a name for the middle state and think it would be better to label the middle state. We added the metal-binding state label in the revised Figure 6 and also added “on the other hand, optimal alignment of a deprotonated water and Mg2+ within the active site, labeled as metal-binding state, leads to irreversible bond breakage (Fig. 6a)” within the text.

(2) The section on DNA hydrolysis assay (Materials and Methods) is not well described. In this section, the authors should summarize the methods for the experiments in Figure 4 AC, Figure 5BC, Figure S3C, Figure S4EF, and Figure S6AB. The authors presented some graphs for the reactions. For clarity, the author should state in the legends which experiments the results are from (in crystallo or in solution). Please check and modify them.

Thank you for the suggestion. We have added four paragraphs to detail the experimental procedures for experiments in these figures. In addition, we have checked all of the figure legends and labeled them as “in crystallo or in solution.” To clarify, we also added “in crystallo” or “solution” in the corresponding panels.

(3) The authors showed the anomalous signals of Mn2+ and Tl+. The authors should mention which wavelength of X-rays was used in the data collections to calculate the anomalous signals.

Thank you for the suggestion. We have included the wavelength of the X-ray in the figure legends that include anomalous maps, which were all determined at an X-ray wavelength of 0.9765 Å.

(4) The full names of "His-Me" and "HNH" are necessary for a wide range of readers.

Thank you for the suggestion. We have included the full nomenclature for His-Me (histidine-metal) nucleases and HNH (histidine-asparagine-histidine) nuclease.

(5) The authors should add the side chain of Arg61 in Figure 1E because it is mentioned in the main text.

Thank you for the suggestion. We have added Arg61 to Figure 1E.

(6) Figure 5D. For clarity, the electron densities should cover the Na+ ion. The same request applies to WatN in Figure S3B.

Thank you for catching this detail. We have added the electron density for the Na+ ion in Figure 5D and WatN in Figure S3B.

(7) At line 269 on page 8, what is "previous H98A I-PpoI structure with Mn2+"? Is the structure 1CYQ? If so, it is a complex with Mg2+.

Thank you for catching this detail. We have edited the text to “previous H98A I-PpoI structure with Mg2+.”

(8) At line 294 on page 9, "and substrate alignment or rotation in MutT (66)." I think "alignment of the substrate and nucleophilic water" is preferred rather than "substrate alignment or rotation".

Thank you for the suggestion. We have edited the text to “alignment of the substrate and nucleophilic water.”

(9) At line 305 on page 9, "Second, (58, 69-71) single metal ion binding is strictly correlated with product formation in all conditions, at different pH and with different mutants (Figure 3a and Supplementary Figure 4a-c) (58)". The references should be cited in the correct positions.

Thank you for catching this typo. We have removed the references.

(10) At line 347 on page 10, "Grown in a buffer that contained (50 g/L glucose, 200 g/L α-lactose, 10% glycerol) for 24 hrs." Is this sentence correct?

Thank you for catching this detail. We have corrected the sentence.

(11) At line 395 on page 11, "The His98Ala I-PpoI crystals of first transferred and incubated in a pre-reaction buffer containing 0.1M MES (pH 6.0), 0.2 M NaCl, 1 mM MgCl2 or MnCl2, and 20% (w/v) PEG3350 for 30 min." In the experiments using this mutant, does a pre-reaction buffer contain MgCl2 or MnCl2?

Thank you for bringing this to our attention. We have performed two sets of experiments: (1) metal ion soaking in 1 mM Mn2+, which is performed similarly as WT and does not have Mn2+ in the pre-reaction buffer; (2) imidazole soaking, 1 mM Mn2+ was included in the pre-reaction buffer. We reasoned that the Mn2+ will not bind or promote reaction with His98Ala I-PpoI, but pre-incubation may help populate Mn2+ within the lattice for better imidazole binding. However, neither Mn2+ nor imidazole were observed. We have added experimental details for both experiments with His98Ala I-PpoI.

(12) In the figure legends of Figure 1, is the Fo-Fc omit map shown in yellow not in green? Please remove (F) in the legends.

We have changed the Fo-Fc map to be shown in violet. We have also removed (f) from the figure legends.

(13) I found descriptions of "MgCl". Please modify them to "MgCl2".

Thank you for catching these details. We have modified all “MgCl” to “MgCl2.”

(14) References 72 and 73 are duplicated.

We have removed the duplicated reference.

**Reviewer #2 (Recommendations For The Authors):**
p. 9, first paragraph, last three lines: "Thus, we suspect that the metal ion may play a crucial role in the chemistry step to stabilize the transition state and reduce the electronegative buildup of DNA, similar to the third metal ion in DNA polymerases and RNaseH." This point is significant but the statement seems a little uncertain. You are saying that the single metal plays the role of two metals in polymerase, in both the ground state and the transition state. I believe the sentence can be stronger and more explicit.

Thank you for raising this point. We suspect the single metal ion in I-PpoI is different from the A-site or B-site metal ion in DNA polymerases and RNaseH, but similar to the third metal ion in DNA polymerases and nucleases. As we stated in the text,

(1) the metal ion in I-PpoI is not required for substrate alignment. The water molecule and substrate can be observed in place even in the presence of the metal ion. In contrast, the A-site or B-site metal ion in DNA polymerases and RNaseH are required for aligning the substrates.

(2) Moreover, the appearance of the metal ion is strictly correlated with product formation, similar as the third metal ion in DNA polymerase and RNaseH.

To emphasize our point, we have revised the sentence as

“Thus, similar to the third metal ion in DNA polymerases and RNaseH, the metal ion in I-PpoI is not required for substrate alignment but is essential for catalysis. We suspect that the single metal ion helps stabilize the transition state and reduce the electronegative buildup of DNA, thereby promoting DNA hydrolysis.”

Minor typos:p. 2, line 4 from bottom: due to the relatively low resolution...

Thank you for catching this. We have edited the text to “due to the relatively low resolution.”

Figure 4F: What is represented by the pink color?

The structures are color-coded as 320 s at pH 6 (violet), 160 s at pH 7 (yellow), and 20 s at pH 8 (green). We have included the color information in figure legend and make the labeling clearer in the panel.

p. 9, first paragraph, last line: ...similar to the third...

Thank you for catching this. We have edited the text.